# Suppressed atmospheric chemical aging of cooking organic aerosol particles in wintertime conditions

Wenli Liu[1], Longkun He[2], Yingjun Liu[2], Keren Liao[2], Qi Chen[2], Mikinori Kuwata[1]

[1]Department of Atmospheric and Oceanic Sciences, Laboratory for Climate and Ocean-Atmosphere Studies, Peking University, Beijing, 100871, China
[2]State Key Joint Laboratory of Environmental Simulation and Pollution Control, BIC-ESAT and IJRC, College of Environmental Sciences and Engineering, Peking University, Beijing, 100871, China

*Correspondence to*: Mikinori Kuwata (kuwata@pku.edu.cn)

**Abstract.**

Cooking organic aerosol (COA) is one of the major constituents of particulate matter in urban areas. COA is oxidized by atmospheric oxidants such as ozone, changing its physical, chemical and toxicological properties. However, atmospheric chemical lifetimes of COA and its tracers such as oleic acid are typically longer than that have been estimated by laboratory studies. We tackled the issue by considering temperature. Namely, we hypothesize that increased viscosity of COA at ambient temperature accounts for its prolonged atmospheric chemical lifetimes in wintertime. Laboratory generated COA particles from cooking oil were exposed to ozone in an aerosol flow tube reactor for the temperature range of -20 °C - 35 °C. The pseudo-second order chemical reaction rate constants ($k_2$) were estimated from the experimental data by assuming a constant ozone concentration in the flow tube. The estimated values of $k_2$ decreased by orders of magnitude for lower temperatures. The temperature dependence in $k_2$ was fit well by considering diffusion-limited chemical reaction mechanism. The result suggests that increased viscosity was likely the key factor to account for the decrease in chemical reactivity at the reduced temperature range, though the idea will still need to be verified by temperature-dependent viscosity data in the future. In combination with the observed global surface temperature, the atmospheric chemical lifetimes of COA were estimated to be much longer in wintertime (>1 hour) than that in summertime (a few minutes) for temperate and boreal regions. Our present study demonstrates that the oxidation lifetimes of COA particles will need to be parameterized as a function of temperature in the future for estimating environmental impacts and fates of this category of particulate matter.

## 1 Introduction

Organic aerosol (OA) is a major component of atmospheric particles (Zhang et al., 2011; Laskin et al., 2019). OA plays important roles in the climate and air quality (Menon et al., 2008; Liu et al., 2013; Robinson et al., 2018; Liu et al., 2019). One of the key sources of OA in urban areas is cooking (Rogge et al., 1991; Hildemann et al., 1991; Crippa et al., 2013; Lee et al., 2015; Sun et al., 2016; Daellenbach et al., 2017; Guo et al., 2020; Huang et al., 2021). Atmospheric abundance of COA is typically estimated by factor analysis of aerosol mass spectra or by employing chemical tracers (Zhang et al., 2011; Huang et al., 2021). Hildemann et al. identified that cooking organic aerosol (COA) accounted for 21 % of mass loading of OA in Los

Angeles, USA (Hildemann et al., 1991). The corresponding values for European cities were reported to be at around 10% (Crippa et al., 2013; Daellenbach et al., 2017). In the case of the Asian region, high abundances of COA have been observed especially in cities in China, including Beijing, Shanghai, Guangzhou, and Hong Kong (Lee et al., 2015; Sun et al., 2016; Guo et al., 2020; Huang et al., 2021). The mass fractions of COA were highly variable (8-33 %) in China, likely due to differences in local source distributions and uncertainties in source apportionment (Zheng et al., 2023; Miao et al., 2021).

Processes involved in COA emissions include meat cooking operations and stir/deep frying by vegetable oils (Zhao et al., 2007; Weitkamp et al., 2008). In both cases, fatty acids such as oleic, linoleic, palmitic, and stearic acids were identified as the major marker compounds for COA (Song et al., 2023; Zhao et al., 2007). As COA contains unsaturated fatty acids such as oleic and linoleic acids, oxidation of COA occurs following chemical reactions with ozone, OH, and $NO_3$ (Weitkamp et al., 2008; Li et al., 2020; Wang and Yu, 2021).

The chemical aging processes of COA and its surrogates change their hygroscopicity and toxicity (Liu et al., 2021; Wang et al., 2021). For instance, ozonolysis of oleic and linoleic acids particles makes them to be highly active as cloud condensation nuclei (CCN) (Broekhuizen et al., 2004; Shilling et al., 2007). Oxidative potential of COA enhances following oxidation (Wang et al., 2021). In addition, COA aging processes need to be well understood for appropriately appointing sources of OA (Reyes-Villegas et al., 2018). It should be noted that the COA factor from factor analysis of the aerosol mass spectrometric measurements may contain both fresh and aged COA, while the profiles of molecular level chemical tracers may change due to atmospheric oxidation (Huang et al., 2021).

The discrepancy in the atmospheric chemical lifetimes of COA/corresponding marker compounds and estimations of the corresponding values by laboratory experiments have been discussed for a few decades (Robinson et al., 2006; Rudich et al., 2007; Weitkamp et al., 2008; Wang and Yu, 2021). Atmospheric chemical lifetimes of oleic and linoleic acids estimated by laboratory experiments were typically in the order of minutes (Morris et al., 2002; Knopf et al., 2005; Hung and Tang, 2010; Liu et al., 2023). The laboratory data for ozonolysis of oleic acid were recently compiled and analyzed using the kinetic multilayer model (Berkemeier et al., 2021). On the other hand, a field observation in Los Angeles, USA suggested that the atmospheric chemical lifetime of oleic acid was on the order of days (Rogge et al., 1991; Zahardis and Petrucci, 2007). Recent observational studies in the Yangtze River Delta region in China suggested that the nighttime chemical lifetimes of oleic acid from cooking in the region was in the order of hours (Wang and Yu, 2021).

The discrepancy among the field and laboratory chemical lifetimes of COA markers has typically been associated with chemical composition of COA (Rudich et al., 2007). Namely, mixing of saturated fatty acids from cooking such as stearic and palmitic acids makes COA to be highly viscous or solid at room temperature, reducing chemical reactivity (Knopf et al., 2005). Chemical reactions of COA with ozone also lead to phase transition of liquid to semi-solid or solid phases, potentially due to the formation of high-molecular-weight chemical species (Xu et al., 2022).

Recent laboratory studies started suggesting that temperature may also be an important factor in regulating chemical reactivity of COA (Li et al., 2020; Kaur Kohli et al., 2023; Liu et al., 2023). Reactive uptake coefficient ($\gamma$) of ozone by canola oil film on a coated flow tube reactor decreased by an order of magnitude at the melting point (Li et al., 2020). Ozonolysis of oleic

acid particles that contained ammonium sulfate seeds became unmeasurably slow by an aerosol flow tube reactor for the temperature range below its melting point (Liu et al., 2023). Considering that ambient temperatures during wintertime in many cities in the world are significantly lower than a typical laboratory condition (20-25 °C), temperature could also play an important role in regulating the atmospheric chemical lifetimes of COA.

Most of previous temperature-dependent oxidation experiments of COA were conducted using organic films or droplets on substrates (Hung and Tang, 2010; Li et al., 2020; Liu et al., 2023). However, the existence of substrates may influence physicochemical processes such as phase transition, impacting chemical reactivity (Hearn and Smith, 2005; Liu et al., 2023; De Gouw and Lovejoy, 1998). Low temperature oxidation experiments for suspending COA particles are still needed.

In this study, we have conducted laboratory experiments for testing the role of temperature on chemical reactivity of COA.

The laboratory study was conducted using an aerosol flow tube for the temperature range of -20 °C to 35 °C. The COA particles for the experiments were produced by heating canola oil, Chinese-style hot pot soup base, and lard. The experimental results were used for evaluating changes in chemical lifetimes of COA as a function of temperature.

## 2 Materials and Methods

Figure 1 shows the experimental setup. The experimental setup consisted of three components, including particle generator,
aerosol flow tube, and online instruments for quantifying the chemical composition and size distribution of aerosol particles. Purified air produced by the zero-air generator (Model 747−30, AADCO Instruments, Inc.) was employed for the setup. Stainless steel, copper and conductive polytetrafluoroethylene (PTFE) tubings were employed for particle flows. Short pieces of Tygon tubings were also used for connecting them. PTFE tubing was employed both for zero air and ozone flows. Table S1 summarizes the list of experimental runs. Further details about the experimental setup can be found in our previous publication
(Liu et al., 2023).

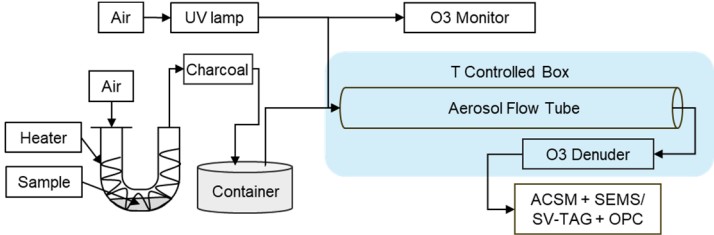

**Figure 1. Schematic diagram of the experimental setup. Particles were measured using the SV-TAG and OPC for Exp. #8, 15 and 22. The ACSM and SEMS were employed for other experiments.**

### 2.1 Generation of COA particles

Three types of cooking materials, including canola oil (product ID: 10057184957713, Jinlongyu Group Co., Ltd.), Chinese-style hot pot soup base (Sichuan spicy hot pot, product ID: 858421, Haidilao) and lard (produced by a cafeteria in the campus

of Peking University), were employed for the experiment. Lard was extracted by heating pork fat using a Chinese frying pan for around half hour following addition of a trace amount of vegetable oil and water. The samples were stored in a refrigerator until usage.

COA particles were generated by heating cooking oils to $180 \pm 2$ °C using a U-tube wrapped with six layers of aluminum foils and a heating tape inside. A thermocouple was installed for measuring the temperature of the outer wall of the U-tube. Approximately 3 ml of samples were put in the bottom of the U-tube. The hot pot soup base contained some solid components, including spices. Only the supernatant liquid was employed for the experiments. A small amount of lard was heated during the sample preparation process, as it formed solid phase at room temperature. The air flow rate passing through the U-tube was
controlled at 2 lpm by a mass flow controller (MFC, Alicat Scientific, Inc.). Nucleation and condensation of the heated samples occurred in an air-cooled condenser. An impactor (Model 8008, Brechtel Manufacturing, Inc.) was used to remove large particles. Volatile organic compounds (VOCs) were removed by a diffusion dryer (AGS-Dryer, Brechtel Manufacturing, Inc.) containing activated charcoal for avoiding formation of secondary organic aerosol particles during the ozonolysis experiments. Particles were stored in a 100 l stainless-steel container for three hours to stabilize size distribution by coagulation prior to
conducting the oxidation experiments. Particle mass concentration in the tank reduced by approximately 50 % following 3 hours of storage due to wall losses. No significant change in particle size distribution was observed after 2 hours (Fig. S1). Mode diameter for the number size distribution in the tank maximumly shifted by 10 % during a set of experiment (Fig. S2).

## 2.2 Temperature-controlled aerosol flow tube

An aerosol flow tube, which was installed in a temperature-controlled box, was employed for the ozonolysis reaction. The
aerosol flow tube consisted of a borosilicate glass tube (5 cm in inner diameter, and 1 m in length) and a movable injector (1/4 inch stainless steel tubing supported by 3/8 inch stainless steel tubing). The positions of the movable injector were changed for investigating oxidation kinetics for the reaction time of 0 to 60 seconds at room temperature. Temperature of the box for the aerosol flow tube was controlled for the range of -20 °C to 35 °C by a chest freezer and a temperature-controlled liquid circulator. The temporal variation of temperature was $\pm 0.2$ °C at $-20$°C.
Ozone was produced using an ultraviolet lamp (UV-M, Beijing Tonglin Technology Co., Ltd). Ozone concentration was continuously monitored by an ozone analyzer (Model 49i, Thermo Fisher Scientific, Inc.). Ozone concentration was adjusted to be 450 ppb and 7 ppm for kinetics and products investigation experiments, respectively. A diffusion dryer containing the ozone destruction catalyst (F800, Beijing Tonglin Technology Co., Ltd) was connected immediately after the flow tube for terminating chemical reactions.

## 2.3 Particle characterization

Chemical composition of particles was monitored using the time-of-flight aerosol chemical speciation monitor (ToF-ACSM, Aerodyne Research, Inc.) equipped with the PM$_{2.5}$ aerodynamic lens and capture vaporizer (Zheng et al., 2020). The range of mass-to-charge ($m/z$) ratio for the mass spectra was 12 to 210. The multilinear engine (ME-2) solver was employed for factor

analysis of the aerosol mass spectra (Budisulistiorini et al., 2021; Liu et al., 2023). A scanning electrical mobility system (SEMS, Brechtel Manufacturing, Inc.), which consisted of the combination of the differential mobility analyzer (DMA) and mixing-based condensation particle counter (MCPC), measured size distribution of particles. Size distributions of particles were recorded every 3 minutes in 60 bins for the diameter range of 10-600 nm.

In addition, the semi-volatile thermal-desorption aerosol gas chromatography (SV-TAG, Aerodyne Research Inc. & Aerosol Dynamic Inc.) was used for experiments at room temperature for molecular level chemical analysis (Li et al., 2022; Liu et al., 2023). Standards for major fatty acids in COA, including linoleic, oleic, palmitic, and stearic acids (Table S2) were employed for calibrating the instrument. The SV-TAG was operated in 'bypass' mode for sampling both gas- and particle-phase species. Different instrumental settings (oven temperature of the concentrating trap and flow rate for the gas chromatography) were employed in experiments for identifying compounds (Fig. 2 and Fig. S3) and for reaction kinetics. Identifications of detected chemical species were conducted using MassHunter (Agilent). An optical aerosol counter (OPC, Model 11-D, GRIMM Aerosol Technik Ainring) was employed together with the SV-TAG for measuring mass concentration.

# 3 Results and discussion

## 3.1 Chemical composition of COA particles

Figure 2 shows the gas chromatogram for COA particles from the hot pot soup base. Chromatograms for all types of COA are shown in Fig. S3, in addition to the corresponding background data. The SV-TAG identified more than 500 chemical species in the COA particles. The number of identified organic molecular formula was dominated by oxygenated compounds such as carboxylic acids. Nitrogen-containing compounds and hydrocarbons were also detected (i.e., amides and alkanes). In addition, there were some chemical species containing sulfur, phosphorus, or halogen atoms. Examples include benzenesulfonamide, phosphinic acid, and bromofluorene. It should be noted that the identification of these chemical species was conducted using the unit-mass resolution mass spectrometer of the SV-TAG. Future employment of the high-resolution mass spectrometer will be needed for accurately identifying these chemical species that contain heteroatoms. Peaks of plasticizers such as phthalate were also identified. The plasticizers were likely originated from contaminants in the zero air in addition to the plastic tubings, as they existed in background measurement (Fig. S3).

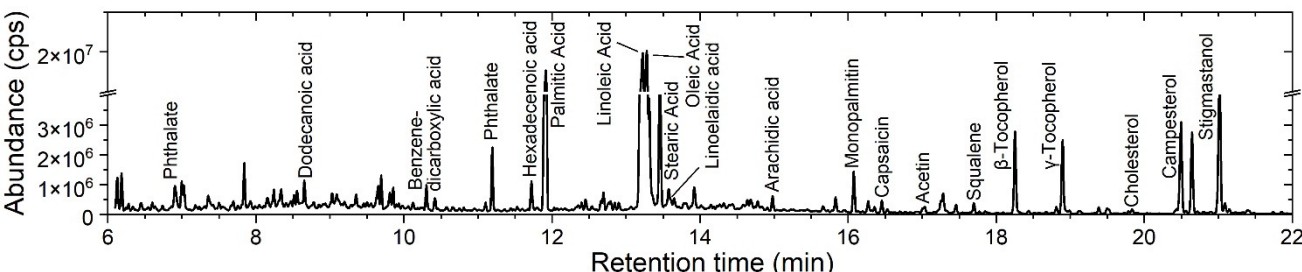

Figure 2: Gas chromatogram of hot pot COA particles prior to ozone exposure.

Oleic, linoleic, palmitic, and stearic acids were the four dominant fatty acids, accounting for around 20 % of the total particle mass. Fatty acids were likely released from thermal degradation of triglycerides and phospholipids (Abdullahi et al., 2013; Zhao et al., 2015). Figure 3 shows the mass fractions of these four fatty acids in investigated COA. Canola oil particles were rich in unsaturated fatty acids (i.e., linoleic and oleic acids), while COA generated from lard contained the highest fraction of saturated fatty acids (i.e., palmitic and stearic acids). High abundance of linoleic acid in COA particles from hot pot might be

originated from chicken oil in the sample (Song et al., 2023). Some other fatty acids with even carbon numbers, such as dodecanoic, myristic, hexadecenoic, linoelaidic, and arachidic acids were also detected, though abundances of signals for these compounds were minor.

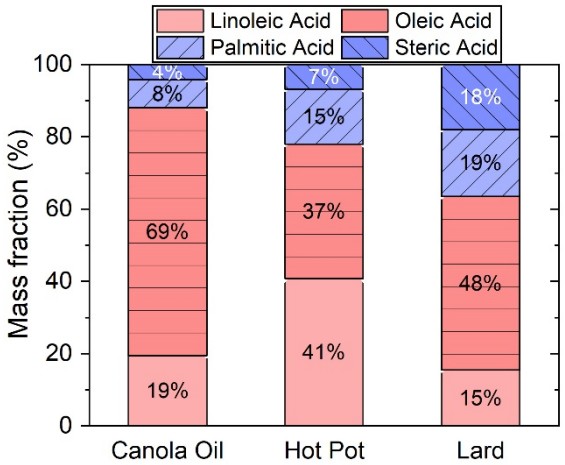

**Figure 3: Mass fractions of four major fatty acids in canola oil, hot pot and lard COA particles.**

Some other minor organic species were also detected, such as sterols, tocopherols, and polycyclic aromatic hydrocarbons (PAHs). Sterols widely exist in vegetable and animal tissues (Sikorski and Kolakowska, 2010).The SV-TAG detected sterols such as cholesterol, campesterol, sitosterol, stigmasterol and fucosterol. The main dietary source of tocopherol is vegetable oil (Jiang et al., 2001). Distinct signals from β, γ, and δ tocopherols were found in three types of COA particles, especially in hot pot particles. Usage of vegetable oil during the extraction process might have contributed to the existence of this compound in

lard COA. PAHs such as benzo(a)pyrene, benz(a)anthracene, pyrene and fluoranthene were detected, although their abundances were limited. Capsaicin was non-negligible only for hot pot COA particles, likely originating from spices.

Figure 4 shows the ToF-ACSM mass spectra of three types of COA particles. The major peaks for the mass spectra included $m/z$ = 41, 55, 67 and 91, as have been reported in previous COA studies (Mohr et al., 2009; He et al., 2010; Kaltsonoudis et al., 2017). Ratios of $f_{55}/f_{57}$ ($f_{55}/f_{57}$ = 2.4-3.2) were consistent with previous COA measurements (Zhang et al., 2011;

Kaltsonoudis et al., 2017). High-molecular weight ions ($m/z$ > 100) that have been observed for oleic acid ($m/z$ =178, 191 and 202) were also detected (Hu et al., 2018; Liu et al., 2023). These dominant peaks were most likely hydrocarbon ions, as have been reported by previous COA studies (He et al., 2010; Kaltsonoudis et al., 2017; Hu et al., 2018).

Generally, mass spectra for the three types of COA were similar ($r^2 > 0.95$). Similarities in aerosol mass spectra for COA from various sources have also previously been reported (He et al., 2010). The mass spectrum of hot pot COA particles exhibited relatively large differences from the other two types of COA. Fractions of signals at $m/z$ = 105, 107, 117, and 137 were 2-5 times higher in hot pot COA particles.

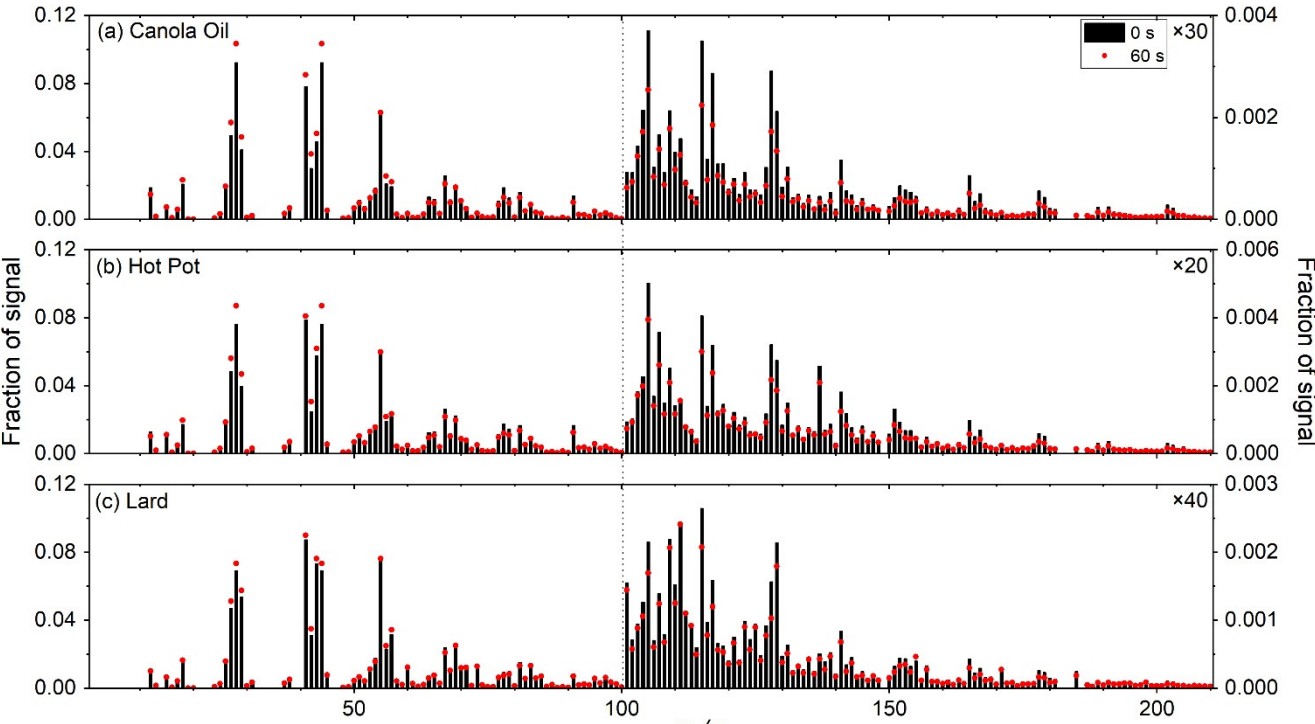

**Figure 4: Mass spectra of (a) canola oil Exp. #1, (b) hot pot Exp. #9 and (c) lard Exp. #20 COA particles (black bars). Mass spectra of particles following $2.7 \times 10^{-5}$ atm s of ozone exposure are shown in red dots. The signal fractions of $m/z$ of 101-210 are enhanced by 30, 20 and 40 times as noted in the figure.**

### 3.2 Chemical characteristics of oxidized COA particles at room temperature

Ozonolysis of COA particles changed their chemical compositions (Fig. 5). Especially, abundances of unsaturated fatty acids decreased following oxidation, as C=C double bonds in unsaturated compounds are highly reactive with ozone (Zahardis and Petrucci, 2007). Abundance of azelaic acid, which formed following ozonolysis of oleic acid, increased following oxidation. These changes were ubiquitously observed for all types of COA. There were no significant changes in mass concentration of saturated fatty acids by oxidation.

The ToF-ACSM data also demonstrated reductions in high molecular weight ions (i.e., $m/z$ =165, 191, and 202) following oxidation, likely due to fragmentations of unsaturated acids by ozonolysis. As a result, the contributions of smaller molecular

ions such as $m/z$ = 43 and 44 increased (Fig. 4). These data were qualitatively in line with the result of the SV-TAG measurements that the oxidation processes induced fragmentation of organic compounds in COA.

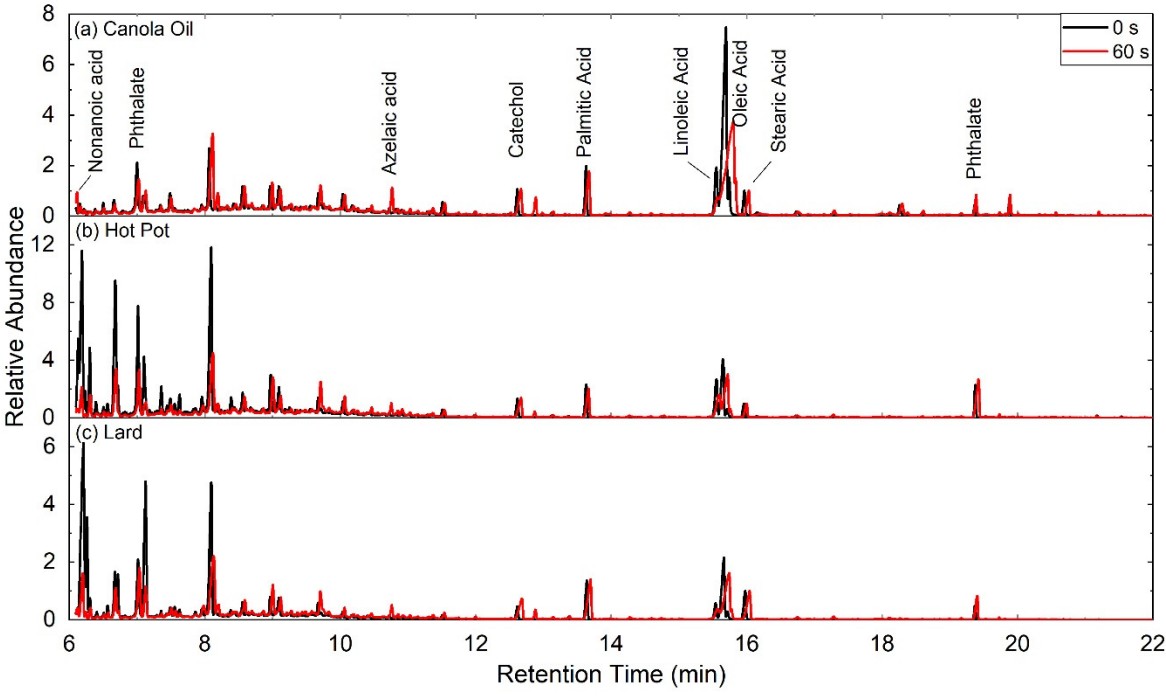

**Figure 5**: Gas chromatograms of (a) canola oil, (b) hot pot and (c) lard COA particles for prior (black) and following $2.7 \times 10^{-5}$ atm s of ozone exposure (red). The abundances are normalized by the intensity of stearic acid for each measurement.

**3.3 Estimation of the pseudo-second order reaction rate constants**

The ME-2 analysis was conducted for quantitatively estimating the mass fraction of COA that had been oxidized using the aerosol mass spectra. The details about the ME-2 analysis are provided in Text S1. Briefly, two-factor solutions were obtained for each set of experiment. A factor that corresponded to COA prior to ozone exposure was denoted as 'fresh COA', while another one was named as 'oxidized COA'. The value for the degree of freedom (*a-value*) of the mass spectra of particles prior

to ozone exposure in the aerosol flow tube was set to be 0.2. The mass spectra range of $m/z$ = 41 to 210 were employed for the analysis. The mass spectra for the ozone exposure of 7 ppm at 25 °C were analyzed together with each experimental dataset to constrain the profile for reaction products. The resulting factor profiles of two factors for each type of COA are summarized in Fig. S4.

Figure S5 shows the change in the fractional contributions of the 'fresh COA' factor ($f_{fresh}$) following ozone exposure for the

lard experiments. The values of $f_{fresh}$ exponentially decreased after COA particles were exposed to ozone. Ozone concentration

was assumed to be a constant value, as an excess amount of ozone was injected to the flow tube. As a result, the process was fit by the following equation by assuming the pseudo-second order reaction with ozone:

$$f_{fresh} = f_{fresh\_0} \, exp(-k_2 P_{O3} t) \tag{1}$$

The term of $f_{fresh\_0}$ indicates the mass fraction of 'fresh COA' prior to ozone exposure. The retrieved value of $f_{fresh\_0}$ was 1.00 ± 0.02 for all the cases, agreeing well with the expectation that 'fresh COA' was the sole component prior to ozone exposure. $P_{O3}$ corresponds to the partial pressure of ozone. The residence time of COA particles in the reactor is $t$. $P_{O3}t$ corresponds to ozone exposure. The value of $k_2$ corresponds to the pseudo-second order reaction rate constant for the chemical reaction of Fresh COA + $O_3$ → Oxidized COA. It should be noted that $f_{fresh}$ was occasionally larger than $f_{fresh\_0}$ when the chemical reaction was extremely slow/negligible at low temperatures. As the ACSM is a mass-based instrument, detecting changes in chemical composition due to ozonolysis is challenging when the reacted mass fraction is small. The output of the ME-2 analysis would have relatively large uncertainties when the change in chemical composition is comparable to or less than fluctuations in experimental data. In these cases, $k_2$ was forced to be zero in the following analysis.

In the case of the SV-TAG data, the oxidation processes of unsaturated fatty acids (i.e., oleic and linoleic acids) were parameterized by employing a saturated fatty acid (stearic acid) as a tracer (Wang and Yu, 2021).

$$\frac{[f_{unsaturated\_fatty\_acid}]}{[f_{stearic\_acid}]} = \frac{[f_{unsaturated\_fatty\_acid\_0}]}{[f_{stearic\_acid\_0}]} \, exp(-k_2 P_{O3} t) \tag{2}$$

In the above equation, $f_{unsaturated\_fatty\_acid\_0}$, $f_{unsaturated\_fatty\_acid}$, $f_{stearic\_acid\_0}$, and $f_{stearic\_acid}$ correspond to abundances of unsaturated fatty acids prior to ozone exposure, unsaturated fatty acids following ozone exposure, stearic acid prior to ozone exposure, stearic acid following ozone exposure, respectively.

The obtained values of $k_2$ for 25 °C are compared in Table 1, along with the corresponding values for oleic acid particles retrieved from our previous study (Liu et al., 2023). Generally, the values of $k_2$ were in the range of 0.05-0.09 ppb$^{-1}$ h$^{-1}$. The values of $k_2$ of COA particles obtained by the ME-2 analysis of the ToF-ACSM data and molecular level measurements of oleic acid by the SV-TAG agreed within the differences of 10-40 %, suggesting that the time scale for chemical aging of COA and oleic acid agreed within the range of uncertainty.

**Table 1. Comparison of obtained values of $k_2$ (ppb$^{-1}$ h$^{-1}$) for oleic acid (OL) in particles by the SV-TAG and whole particles by the ACSM at 25 °C.**

| Type | $k_2$ for OL in particles (by SV-TAG) | $k_2$ for whole particles (by ACSM) |
|---|---|---|
| Oleic Acid [a] | 0.067 | 0.078 ± 0.008 |
| Canola Oil | 0.053 | 0.086 ± 0.005 |
| Hot Pot | 0.055 | 0.061 ± 0.003 |
| Lard | 0.049 | 0.056 ± 0.003 |

ª Retrieved from our previous study (Liu et al., 2023)

The similarities in the values of $k_2$ by the ME-2 analysis and SV-TAG were reasonable, considering that unsaturated fatty acids were the major chemical species that were responsible for the oxidation processes. Oxidation of linoleic acid is known to be 1.6 times faster than that of oleic acid (Moise and Rudich, 2002; Hearn and Smith, 2004). However, it only occupied a small

portion, except for hot pot COA particles. A previous study for reactive uptake coefficient of ozone by canola oil film ($\gamma = 0.6 \times 10^{-3}$) (Li et al., 2020) also reported that the chemical reaction process was comparable to that by oleic acid ($\gamma = 0.8 \times 10^{-3}$) (Thornberry and Abbatt, 2004). Differences in experimental procedures could also account for the discrepancy between the two studies.

There were some variabilities in particle number size distributions among the experiments. The mode diameters for the COA

particles were 300-400 nm (Fig. S6), while the corresponding values for oleic acid particles were at around 400 nm. The size ranges were comparable to the ambient COA particles in Beijing (Ma et al., 2023). The differences in reactive uptake coefficients for oleic acid particles would change by less than 5 % for 200 and 400 nm particles, leading to negligibly small changes in $k_2$ values (approximately 10 %) (Morris et al., 2002; Smith et al., 2002). The variabilities in particle sizes among the experiments did not likely affect the experimental results of the present study.

**3.4 Temperature dependence in $k_2$**

Figure 6 summarizes the obtained values of $k_2$ plotted against temperature. In general, $k_2$ decreased systematically for lower temperatures. On the other hand, the values of $k_2$ for elevated temperature ranges ($T > 30$ °C) were higher than that for room temperature ($T \sim 25$ °C). For the temperature range above 25 °C, the retrieved values of $k_2$ for all types of COA and oleic acid particles were only different by less than 3 times. For -10 °C $< T <$ 10 °C, the values of $k_2$ were different by more than an order

of magnitude, depending on COA types. All types of COA were not highly reactive for $T <$ -15 °C.

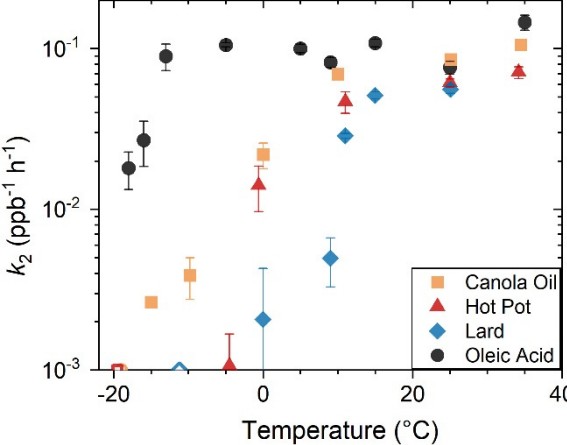

**Figure 6: Dependence of $k_2$ on temperature. Data for oleic acid were retrieved from our previous study (Liu et al., 2023). The values of $k_2$ for Exp. #7, 10 and 21 were unmeasurably slow. Thus, the corresponding data are shown as open symbols at the bottom.**

The change of $k_2$ with temperature was gradual, rather than abrupt (Fig. 6 and Fig. S7). The gradual reduction in chemical reactions for reduced temperature range has previously been reported for organic films that experienced glass transitions using coated flow tube experimental setup (Li et al., 2020). The transition temperature was defined as the point at which $k_2$ became an order of magnitude smaller than that at room temperature. The transition temperatures were at around -15 °C (canola oil), -5 °C (hot pot), and 10 °C (lard). Below these temperatures, the values of $k_2$ were not highly sensitive to temperature/unmeasurably small. In the case of chemical systems that experienced glass transition, similar changes in γ were also observed using the coated flow tube (Li et al., 2020; Moise and Rudich, 2002). Namely, γ was almost constant for the temperature range below the glass transition temperature, as surface reactions dominated for the region. However, γ increased almost linearly with temperature for the warmer region. A similar transition was also likely occurring to the COA particles. COA particles that contained higher fractions of saturated fatty acids had smaller $k_2$ for a certain temperature, as well as higher transition temperature. Previous laboratory studies demonstrated that mixing of saturated fatty acids such as lauric, myristic, stearic, and palmitic acids reduced chemical reactivity of oleic acid particles at room temperature (Hearn and Smith, 2005; Katrib et al., 2005; Knopf et al., 2005). We hypothesize that enhanced abundance of saturated fatty acids increased transition temperatures of COA particles, though further studies that directly quantify viscosity or glass transition would still be needed for confirming the idea. Previous studies demonstrated that numerous processes such as gas and bulk phase diffusion, adsorption and desorption, surface and bulk reaction are involved in determining oxidation rate of aerosol particles (Berkemeier et al., 2021; Pöschl et al., 2007; Li and Knopf, 2021; Willis and Wilson, 2022). Especially, accurate estimation of viscosity is important. A few methods were established to estimate the viscosities and glass transition points of organic compounds based on the chemical compositions and elemental ratios (Derieux et al., 2018; Ceriani et al., 2007). Simultaneous measurements of chemical composition and reactivity will be needed for understanding and estimating chemical aging time scale using molecular-level information of COA particles.

## 4 Atmospheric implications

Our laboratory experiments in the present study demonstrated that pseudo-second order reaction rate constants $k_2$ between COA and ozone were highly sensitive to temperature. Li et al (2020) reported that the reaction of ozone and canola oil liquid film was not gas diffusion limited, meaning that the determining factor for $k_2$ should be identified in particle phase. For estimating the potential impact of temperature on atmospheric chemical lifetimes of COA, the experimentally obtained $k_2$-$T$ relationships were phenomenologically fit by equations. The relationships do not follow the Arrhenius equation well (Fig. S8), suggesting that the major factor that influences the $k_2$-$T$ relationships is not related to the activation energy of the chemical reaction. In the case of diffusion-controlled chemical reaction processes, the overall reaction rate constant is known to be

related to viscosity (η) (Krenos, 2001). The temperature dependence in η can be evaluated by the Vogel–Fulcher–Tammann (VFT) equation;

$$\ln \eta = A + \frac{B}{T-T_0} \tag{3}$$

In the above equation, $A$, $B$, and $T_0$ are empirically obtained parameters. By considering the VFT equation, $k_2$ values were fit by the following equation under the assumption that the oxidation processes of COA were controlled by viscosity change.

$$\ln k_2 = \alpha_1 + \frac{\alpha_2}{T+\alpha_3} \tag{4}$$

In the above equation, $\alpha_1$, $\alpha_2$, and $\alpha_3$ are parameters that are empirically optimized. The fitting results for the three types of laboratory generated COA are summarized in Fig. S9. The corresponding optimized parameters are tabulated in Table S3. VFT equation fit the experimental data well, demonstrating that the bulk diffusion was likely the key factor in controlling the reaction rate of COA particles.

Figure 7 shows the distributions of estimated atmospheric chemical lifetimes of COA with respect to ozonolysis in June and December. The figure was created by assuming that chemical reactivity of ambient COA was comparable to that from canola oil. Monthly mean surface air temperature data were obtained from the website of the Physical Sciences Laboratory of NOAA (https://psl.noaa.gov/mddb2/makePlot.html?variableID=1603) for June and December of 2021 (Fig. S10). Ozone concentration was assumed to be 30 ppb. Qualitatively similar estimations were also obtained for other types of COA (Fig. S11) and for other ozone concentrations (Fig. S12).

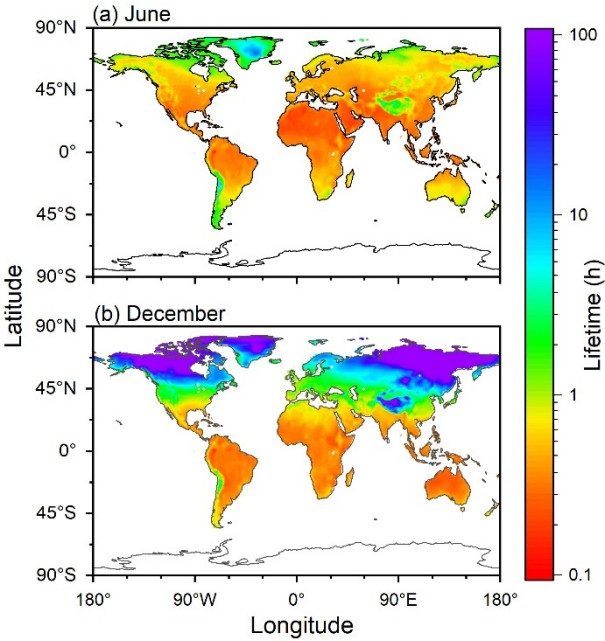

**Figure 7: Estimated chemical lifetimes of COA during (a) June and (b) December. The fit function for the experimental data of canola oil COA was employed for the calculation by assuming the ozone concentration of 30 ppb.**

Temperature was between 10-30 °C for most areas in June, except for Greenland and the Tibetan plateau. As $k_2$ only changed by less than twice for the temperature range, the estimated atmospheric chemical lifetimes were shorter than one hour and did not exhibit an extensive variation.

In December, temperatures of the tropical and southern hemisphere regions were high. Temperatures in high-latitude regions in the northern hemisphere were lower than 0 °C, resulting in atmospheric chemical lifetimes of longer than 10 hours. In the case of the mid-latitude regions, COA may survive in the atmosphere for the time scale of 1-10 hours, consistent with recent observational studies in some cities in China (Wang and Yu, 2021; Li et al., 2023).

The above discussion only considered chemical reactions with ozone under dry conditions, while COA particles were also highly reactive with other atmospheric oxidants such as OH and $NO_3$ radicals (Li et al., 2020). In addition, the discussion ignored changes in chemical reactivity of COA particles that might occur following initial reactions (Hosny et al., 2016; Berkemeier et al., 2021). Potential influence of hygroscopic growth of partially aged COA on its chemical reactivity was also ignored. Mixing state of COA particles that could influence chemical reactivity at low temperature was also not considered (Liu et al., 2023; Ma et al., 2023).

As atmospheric chemical lifetimes of oleic and linoleic acids are short, online measurement techniques are required for estimating their atmospheric chemical lifetimes from ambient data. So far, such studies have only been conducted in a limited number of cities. Further observational evidence will be needed for evaluating the roles of environmental conditions such as temperature on atmospheric chemical lifetimes of COA. Especially, wintertime observations in cold regions would be important. We suggest that future laboratory and field studies of COA will warrant underpinning its atmospheric fate and environmental impacts.

**5 Conclusions**

We conducted a series of laboratory experiments to investigate the influence of temperature on chemical reactivity of COA. Three types of COA particles were generated by heating canola oil, hot pot soup base, and lard. The produced COA particles were exposed to ozone in an aerosol flow tube reactor under the temperature range of -20 °C - 35 °C. Changes in chemical composition of COA particles were characterized by the ToF-ACSM and SV-TAG. The experimental results were used for calculating the chemical reaction rate constants $k_2$. The observed temperature dependences in $k_2$ were fit well by the VFT-equation, suggesting that reduced viscosity was the key factor to account for the decrease in chemical reactivity. The parameterized temperature-dependent $k_2$ values were used for estimating the chemical lifetimes of COA particles on the global scale, in combination with observed global surface temperature data. During summertime, atmospheric lifetimes of COA are short (< 1 hour) for most regions, while the corresponding value is longer than 10 hours in high-latitude regions in the northern hemisphere during wintertime. This study demonstrates suppressed chemical reactivity of COA particles under low temperatures needs to be considered in simulating COA fate.

## Data availability

The data in this study are available from the corresponding author upon request.

## Author contributions

Conceptualization - MK and WL; Experiment investigation – WL, LH and KL; Data analysis – WL, LH and MK; Resources and Funding acquisition – MK, YL and QC; Writing – original draft preparation – WL and MK; Writing – review & editing – LH, YL, QC.

## Competing interests

One of the co-authors is a member of the editorial board of Atmospheric Chemistry and Physics.

## Acknowledgments

We acknowledge Mr. Zhao at Jiayuan Canteen in Peking University for providing the lard sample for the study. We also thank Shuyan Xing for helping with the TOC figure.

## Financial support

The study was funded by the National Natural Science Foundation of China (42175121, 42150610485, and 92044303).

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
