# Peer review of "Suppressed atmospheric chemical aging of cooking organic aerosol particles in wintertime conditions"

_EGUsphere, 2023_

## Author Comment (AC1)

**Reply to the comment of Referee 1**

We appreciate the editor and reviewers for providing useful comments on revising our manuscript. In the following, the original reviewer comments are shown in black, our responses to the comments are shown in blue, and corresponding updates on the manuscript are represented by *italic* fonts.

This is the review of the manuscript entitled "Suppressed atmospheric chemical aging of cooking organic aerosol particles in wintertime conditions" by Liu et al. This study investigates the chemical lifetime of cooking organic aerosol (COA) of three typical sources including canola oil, hot pot soup, and lard upon exposure to ozone. The second order reaction rate constants of ozone reacting with COA particles were determined as a function of temperature between -20 to 35 C. The reaction kinetics were derived by monitoring the degradation of the condensed phase organic by means of mass spectrometry and gas chromatography. It is observed that the reaction rate decreases by orders of magnitude as the temperature decreases. The temperature dependence of the reaction rate was fitted using a Vogel–Fulcher–Tammann (VFT) equation. This in turn was applied to predict the chemical lifetimes of COA across the globe for the months June and December. During northern hemispheric winter the chemical lifetimes of COA increased significantly.

This work is in the tradition of previous laboratory heterogeneous chemical reaction studies and thus fits the scope and audience of Atmospheric Chemistry and Physics. The experimental approach appears to be sound and the results present novel data. I have a few suggested minor revisions the authors should address before publication of this manuscript. Those points revolve around providing a few more details on the experimental approach and on data interpretation.

We appreciate the reviewer for the recognition of the worth of our research and helpful comments. The concerns raised by the reviewer are addressed in the revised manuscript, as detailed below.

Minor comments:
1. Line 16: In heterogeneous reaction kinetics experiments typically the pseudo-first order decay is monitored. From this the second order rate constant could be derived. Do you imply a pseudo-second-order model that considers chemisorption as the rate-limiting mechanism of the process? Please elaborate.

We thank the reviewer for this comment. As the reviewer mentioned, pseudo-first-order reaction rate constants could be estimated by fitting experimental results using an exponential function. The pseudo-second-order constants were calculated by dividing the first-order rate constant by the ozone concentration during the experiment. As the excess amount of ozone was employed for the experiment, changes in ozone concentration due to the chemical reactions with COA particles were ignorable, meaning that COA particles were exposed to a constant concentration of ozone during an experiment. We clarified this idea in the revised manuscript.

*Line 15: The pseudo-second order chemical reaction rate constants ($k_2$) were estimated from the experimental data by assuming a constant ozone concentration in the flow tube.*

*Line 207: Ozone concentration was assumed to be a constant value, as an excess amount of ozone was injected to the flow tube. As a result, the process was fit by the following equation by assuming the pseudo-second order reaction with ozone:*

2. Line 17: I am not sure if the statement of diffusion limitation is correct in this instance. If a reaction is diffusion-limited then the observed degradation does not reflect the actual reaction kinetics. However, changes in the reaction kinetics with temperature are observed. The authors likely meant to express that the second order rate constant is controlled by diffusion? As outlined further below, I would argue that a fit using a similar equation as the VFT description of viscosity is not a sufficient proof that only diffusion governed the observed temperature dependency of the reactivity.

We appreciate the reviewer for this comment. We agree with the reviewer that the influence of viscosity on reaction kinetics is only inferred, rather than being evidenced by directly monitoring the diffusion process of ozone. We think that future viscosity measurements of COA particles will be needed for experimentally verifying how oxidation kinetics is regulated. We revised this sentence in a more accurate way.

*Line 18: The result suggests that increased viscosity was likely the key factor to account for the decrease in chemical reactivity at the reduced temperature range, though the idea will still need to be verified by temperature-dependent viscosity data in the future.*

3. Line 66: "However,…". I do not understand this statement. If experiments are done correctly, reactive uptake measurements using aerosol particles or films result in the same reaction kinetics (Ammann et al., 2013). There are advantages and disadvantages for both approaches, e.g., gas-phase diffusion limitations. If the aim is to indicate that OA can remain in a metastable liquid phase, i.e., being supercooled (see, e.g., (Hearn and Smith, 2005)), which is less likely to occur using a film, then this has to be more clearly stated. Furthermore, I think it would be fair to acknowledge the study by (De Gouw and Lovejoy, 1998).

We thank the reviewer for this comment. As the reviewer mentioned, experimental approaches that employ substrates were highly prone to induce artifacts especially when the investigated chemical system was supercooled. We clarified it in the main text. In addition, we thank the reviewer for these references.

*Line 70: Most of previous temperature-dependent oxidation experiments of COA were conducted using organic films or droplets on substrates (Hung and Tang, 2010; Li et al., 2020; Liu et al., 2023). However, the existence of substrates may influence physicochemical processes such as phase transition, impacting chemical reactivity (Hearn and Smith, 2005; Liu et al., 2023; De Gouw and Lovejoy, 1998). Low temperature oxidation experiments for suspending COA particles are still needed.*

4. Line 109: I would not call those concentrations "normal" and "high". Both are unrealistically high. Please rephrase. You might want to express those concentrations also as a typically background and urban polluted ozone exposure time.

We thank the reviewer for pointing it out. We revised it as 'kinetics and products investigation experiments' in the revised manuscript.

*Line 116: Ozone concentration was adjusted to be 450 ppb and 7 ppm for kinetics and*

*products investigation experiments, respectively.*

5. Line 113 and following: When determining the heterogeneous oxidation kinetics using aerosol particles, one has to pay attention to how reactivity scales with particle size (surface/volume). See e.g., studies by (Lim et al., 2017; Slade and Knopf, 2014). Have those experiments been conducted? How does the size distribution change prior to and after ozone exposure?

We appreciate the reviewer for this comment. We did not conduct experiments to investigate size dependence in reaction rates. As shown in Figure S6 of the revised manuscript, the mode diameter of particle number size distribution was stable within the range of 300 ± 50 nm, except for a few exceptional cases. The uncertainty in size would induce less than 10 % of errors in $k_2$ according to previous oleic acid ozonolysis researches (Morris et al.2002, Smith et al. 2002). No obvious change was found in size distribution after oxidation except for the 7 ppm experiment.

We added the following paragraph to describe the size distribution of particles and the potential influences on reaction rate constants.

*Line 240: There were some variabilities in particle number size distributions among the experiments. The mode diameters for the COA particles were 300-400 nm (Fig. S6), while the corresponding values for oleic acid particles were at around 400 nm. The size ranges were comparable to the ambient COA particles in Beijing (Ma et al., 2023). The differences in reactive uptake coefficients for oleic acid particles would change by less than 5 % for 200 and 400 nm particles, leading to negligibly small changes in $k_2$ values (approximately 10 %) (Morris et al., 2002; Smith et al., 2002). The variabilities in particle sizes among the experiments did not likely affect the experimental results of the present study.*

*Line 107: Mode diameter for the number size distribution in the tank maximumly shifted 10 % during a set of experiment (Fig. S2).*

6. Somewhere in the introduction, to elevate the discussion, recent modeling studies that account for viscosity changes in multiphase chemical kinetics, could be briefly mentioned. E.g., (Berkemeier et al., 2021).

We appreciate this suggestion from the reviewer. We added the following sentence to address the point.

*Line 53: The laboratory data for ozonolysis of oleic acid were recently compiled and analyzed using the kinetic multilayer model (Berkemeier et al., 2021).*

7. Section 3.3 and Table 1: I struggle to understand Table 1 and suggest elaborating this discussion more. When just quickly looking at the table, its meaning is not very clear. In the second column $k_2$ is derived for only the oleic acid component in the types of particles given in column 1? Whereas column three reflects the reaction kinetics using a wider range of the mass spectrum. Maybe change the table or its description to make this clearer. Except for the value of the previous study, the data is derived from the same experiments?

We thank the reviewer for pointing out this unclear expression. The values in column 2 were obtained from SV-TAG, while these in column 3 obtained from ACSM. Measurements by SV-TAG (column 2) and ACSM (column 3) were done in independent experiments under the same condition. We have updated it in

the revised version.

*Line 230:*

*Table 1. Comparison of obtained values of $k_2$ (ppb$^{-1}$ h$^{-1}$) for oleic acid (OL) in particles by the SV-TAG and whole particles by the ACSM at 25 °C.*

| Type | $k_2$ for OL in particles (by SV-TAG) | $k_2$ for whole particles (by ACSM) |
|------|----------------------------------------|--------------------------------------|

8. Does the difference in particle size distribution among the different aerosol source types matter when comparing their kinetics (Fig. S1)? See also comment above.

We appreciate the reviewer for this comment. We agree particle size distribution is one important factor which influence chemical reaction rates for aerosol particles. As we mentioned in the response to comment #5, difference of 100 nm in particle diameter would not change $k_2$ value by more than 10 %. Therefore, chemical composition and viscosity change of particles were more important in our experiments. We add more description about the influence of particle size of $k_2$ in the revised version.

*Line 240: There were some variabilities in particle number size distributions among the experiments. The mode diameters for the COA particles were 300-400 nm (Fig. S6), while the corresponding values for oleic acid particles were at around 400 nm. The size ranges were comparable to the ambient COA particles in Beijing (Ma et al., 2023). The differences in reactive uptake coefficients for oleic acid particles would change by less than 5 % for 200 and 400 nm particles, leading to negligibly small changes in $k_2$ values (approximately 10 %) (Morris et al., 2002; Smith et al., 2002). The variabilities in particle sizes among the experiments did not likely affect the experimental results of the present study.*

9. It may be worthwhile to mention that you are likely not gas-phase diffusion limited in the case of ozone uptake? I assume the uptake is sufficiently slow. Citing previous studies using canola oil or oleic acid might be helpful in this regard.

We thank the reviewer for this suggestion. We mentioned it and added corresponding references in the revised version.

*Line 279: Li et al (2020) reported that the reaction of ozone and canola oil liquid film was not gas diffusion limited, meaning that the determining factor for $k_2$ should be identified in particle phase.*

10. Line 197-198: Looking at Fig. S5 it seems the ratio was greater one for lowest temperature measurements. Could it be that surface-dominated oxidation resulted in more products that did not volatilize due to lower temperatures?

We thank the reviewer for this question. We agree with the reviewer that surface chemical reactions might be the dominant under low temperature because of enhanced viscosity. However, due to the technical limitation of the ME-2 approach, we are unable to tell relatively small changes in chemical compositions that could be induced by surface reactions. Both the ME-2 and ACSM analysis are mass-based. Thus, the approach is sensitive to chemical reactions in the bulk phase, rather than a surface layer. The following statement was provided in the revised manuscript to address the issue.

*Line 214: It should be noted that $f_{fresh}$ was occasionally larger than $f_{fresh\_0}$ when the chemical reaction was extremely slow/negligible at low temperatures. As the ACSM is*

*a mass-based instrument, detecting changes in chemical composition due to ozonolysis is challenging when the reacted mass fraction is small. The output of the ME-2 analysis would have relatively large uncertainties when the change in chemical composition is comparable to or less than fluctuations in experimental data. In these cases, $k_2$ was forced to be zero in the following analysis.*

11. Line 244: Which transition (phase?) do you mean here?

We thank the reviewer for this comment. We provided the definition of 'transition temperature' in the revised version.

*Line 258: The transition temperature was defined as the point at which $k_2$ became an order of magnitude smaller than that at room temperature.*

12. Line 256-260: Condensed-phase diffusion is related to viscosity. However, I doubt, just because you can fit observations reasonably well with a VFT equation, though fit parameters are arbitrary and have no physical meaning, you can infer that only diffusion controls the entire oxidation process. This comes back to my comment in the abstract. There could be several processes going on in series or parallel which you are not resolving. See, e.g., (Pöschl et al., 2007; Berkemeier et al., 2021; Li and Knopf, 2021; Willis and Wilson, 2022). Clearly, your results demonstrate the importance of bulk diffusion but as long we cannot resolve all the intermediate steps, I suggest stating this observation more conservatively.

We appreciate the reviewer for this suggestion. We agree that many processes controlled the reaction at the same time, while diffusion was the key factor in our experiments. We thank the reviewer for providing the useful references. We updated this point in the revised version.

*Line 270: Previous studies demonstrated that numerous processes such as gas and bulk phase diffusion, adsorption and desorption, surface and bulk reaction are involved in determining oxidation rate of aerosol particles (Berkemeier et al., 2021; Pöschl et al., 2007; Li and Knopf, 2021; Willis and Wilson, 2022). Especially, accurate estimation of viscosity is important.*

*Line 293: VFT equation fit the experimental data well, demonstrating that the bulk diffusion was likely the key factor in controlling the reaction rate of COA particles.*

Technical corrections:

13. Line 121: Omit "also" since you already used "In addition,…".

We have revised it as suggested.

14. Figure 3: Typo in legend "Palmitic".

We are sorry for the typo. We have revised it.

15. Line 154: "species" should be "spices"?

We are sorry for the typo. We have revised it.

16. Line 205: Missing "respectively"?

We are sorry for the missing. We have added it.

References

Ammann, M., Cox, R. A., Crowley, J. N., Jenkin, M. E., Mellouki, A., Rossi, M. J., Troe, J., and Wallington, T. J.: Evaluated kinetic and photochemical data for atmospheric chemistry: Volume VI - heterogeneous reactions with liquid substrates, Atmos. Chem. Phys., 13, 8045-8228, 10.5194/acp-13-

8045-2013, 2013.

Berkemeier, T., Mishra, A., Mattei, C., Huisman, A. J., Krieger, U. K., and Poschl, U.: Ozonolysis of Oleic Acid Aerosol Revisited: Multiphase Chemical Kinetics and Reaction Mechanisms, ACS Earth Space Chem., 5, 3313-3323, 10.1021/acsearthspacechem.1c00232, 2021.

de Gouw, J. A. and Lovejoy, E. R.: Reactive uptake of ozone by liquid organic compounds, Geophys. Res. Lett., 25, 931-934, 1998.

Hearn, J. D. and Smith, G. D.: Measuring rates of reaction in supercooled organic particles with implications for atmospheric aerosol, Phys. Chem. Chem. Phys., 7, 2549-2551, 10.1039/b506424d, 2005.

Li, J. and Knopf, D. A.: Representation of Multiphase OH Oxidation of Amorphous Organic Aerosol for Tropospheric Conditions, Environ. Sci. Technol., 55, 7266-7275, 10.1021/acs.est.0c07668, 2021.

Lim, C. Y., Browne, E. C., Sugrue, R. A., and Kroll, J. H.: Rapid heterogeneous oxidation of organic coatings on submicron aerosols, Geophys. Res. Lett., 44, 949–2957, 10.1002/2017GL072585, 2017.

Pöschl, U., Rudich, Y., and Ammann, M.: Kinetic model framework for aerosol and cloud surface chemistry and gas-particle interactions - Part 1: General equations, parameters, and terminology, Atmos. Chem. Phys., 7, 5989-6023, 2007.

Slade, J. H. and Knopf, D. A.: Multiphase OH oxidation kinetics of organic aerosol: The role of particle phase state and relative humidity, Geophys. Res. Lett., 41, 5297-5306, 10.1002/2014gl060582, 2014.

Willis, M. D. and Wilson, K. R.: Coupled Interfacial and Bulk Kinetics Govern the Timescales of Multiphase Ozonolysis Reactions, J. Phys. Chem. A, 126, 4991–5010, 10.1021/acs.jpca.2c03059, 2022.

---

## Author Comment (AC2)

**Reply to the comment of Referee 2**

We appreciate the editor and reviewers for providing useful comments on revising our manuscript. In the following, the original reviewer comments are shown in black, our responses to the comments are shown in blue, and corresponding updates on the manuscript are represented by *italic* fonts.

In their manuscript, Liu et al. investigated the atmospheric lifetime of cooking organic aerosol across the temperature range of -20 to 35C. They performed laboratory experiments with three types of cooking aerosol precursors: canola oil, hot pot base, and lard. Precursors were heated and emissions were passed through an aerosol flow tube containing ozone. Through SV-TAG and ACSM measurements, they measured changes in aerosol composition upon ozone exposure at different temperatures and then estimated pseudo-second order rate constants as a function of reactor temperature. They suggested that changes to aerosol viscosity at lower wintertime temperatures may increase the chemical lifetime of COA to upwards of 1 hour, while in the summer these aerosols may persist for only a few minutes. I enjoyed reading this paper and I think it fits well within the scope of Atmospheric Chemistry and Physics. I have a few minor comments and requests for clarification.

We thank the reviewer for acknowledging the worth of our work and providing many helpful suggestions. Below are our responses and revisions to these comments.

1.  Line 18: Do you mean "increased" viscosity?

We are sorry for the typo. We have revised it to 'increased viscosity'.

*Line 18: The result suggests that increased viscosity was likely the key factor to account for the decrease in chemical reactivity at the reduced temperature range, though the idea will still need to be verified by temperature-dependent viscosity data in the future.*

2.  Line 30: How much did COA contribute in these European cities that are cited? Would be interesting to add some quantitative information here, as you did for the Los Angeles example.

We thank the reviewer for this suggestion. We added the corresponding quantitative information for European cities in the revised manuscript.

*Line 32: The corresponding values for European cities were reported to be at around 10% (Crippa et al., 2013; Daellenbach et al., 2017).*

3.  Line 33: Do you mean that the mass fractions of COA are highly variable in the Chinese cities you mentioned? If so, please clarify. Also, does the range 8-33% come from the papers cited in the line above?

We thank the reviewer for this comment. The values were obtained from previous observations in some cities in China that were mentioned in the references in the end of this sentence. We have clarified it.

*Line 35: The mass fractions of COA were highly variable (8-33 %) in China, likely due to differences in local source distributions and uncertainties in source apportionment (Zheng et al., 2023; Miao et al., 2021).*

4.  Line 52: "Much longer than the time scale" needs to be quantified; which

time scale? I assume you are referring to "in the order of a few minutes" from the prior sentence? Does this account for both daytime and nighttime chemistry?

We thank the reviewer for this question. The specific value for lifetime of oleic acid was not clearly mentioned in the paper by Rogge et al. It could be calculated as a few days, based on the description in the paper. The time scale likely included both daytime and nighttime chemistry. We have clarified it in the revised manuscript.

*Line 54: On the other hand, a field observation in Los Angeles, USA suggested that the atmospheric chemical lifetime of oleic acid was on the order of days (Rogge et al., 1991; Zahardis and Petrucci, 2007).*

5. Line 97: Can you describe any possible particle losses in the stainless-steel storage container? Three hours seems like plenty of time for losses to occur. Is there mixing in the container during this time?

We appreciate the reviewer for the comments about particle size. The experiment was always conducted with an excess amount of ozone. For instance, 8 ppb of ozone would be needed for completely oxidizing 100ug m$^{-3}$ of oleic acid particles. Our experimental condition (450 ppb of ozone, 50-200 ug m$^{-3}$ of COA) suggests that the experiments were conducted with an excess amount of ozone. Thus, the loss of particles in the stainless-steel container does not influence the experimental result. The point was clarified in the revised manuscript in the following way.

What was the size distribution of particles prior to entering the storage container for coagulation, and why was growing the particles necessary prior to ozonolysis experiments? Just to ensure you were comparable to COA particles in Beijing (mentioned on the next line)?

The container was employed for stabilizing particle size distribution by coagulation, as described in our previous study. We added Figure S1 to demonstrate measured particle size distributions following the storage in the container. The following sentence was provided in the revised manuscript to address the point.

And finally, do the size distributions in figure S1 account for all experiments? If so, the setup looks to be quite reproducible, which is great to see, but I suggest clarifying this.

We produced particles by employing the same protocol for all the experiments. As a result, particle size distributions were similar for each type of aerosol among all the experiments. We added Figure S6 to demonstrate the size distributions in each experiment.

Can you comment on whether the difference in size distribution between COA types could be contributing to differences in reactivity between the COA types?

We appreciate the reviewer for this comment. We agree particle size distribution is one important factor which influence chemical reaction rates for aerosol particles. As shown in Figure S6 of the revised manuscript, the mode diameter of particle number size distribution was stable within the range of 300 ± 50 nm, except for a few exceptional cases. The uncertainty in size would induce less than 10 % of errors in $k_2$ according to previous oleic acid ozonolysis researches (Morris et al.2002, Smith et al. 2002). Therefore, chemical composition and viscosity change of particles were more important in our experiments. We add

*more description about the influence of particle size of $k_2$ in the revised version.*

*Line 104: Particles were stored in a 100 l stainless-steel container for three hours to stabilize size distribution by coagulation prior to conducting the oxidation experiments. Particle mass concentration in the tank reduced by approximately 50 % following 3 hours of storage due to wall losses. No significant change in particle size distribution was observed after 2 hours (Fig. S1).*

*Line 240: There were some variabilities in particle number size distributions among the experiments. The mode diameters for the COA particles were 300-400 nm (Fig. S6), while the corresponding values for oleic acid particles were at around 400 nm. The size ranges were comparable to the ambient COA particles in Beijing (Ma et al., 2023). The differences in reactive uptake coefficients for oleic acid particles would change by less than 5 % for 200 and 400 nm particles, leading to negligibly small changes in $k_2$ values (approximately 10 %) (Morris et al., 2002; Smith et al., 2002). The variabilities in particle sizes among the experiments did not likely affect the experimental results of the present study.*

6. Line 108: 450 ppb is still a very high ozone exposure. I suggest re-phrasing how you label 450 ppb as "normal" ozone levels.

We thank the reviewer for this comment. We revised it as 'kinetics investigation experiments' in the manuscript.

*Line 116: Ozone concentration was adjusted to be 450 ppb and 7 ppm for kinetics and products investigation experiments, respectively.*

7. Line 134: Can you comment on the source of the halogens?

We appreciate the reviewer for pointing this out. The hop pot soup contained seasonings and salt. We suspect that they served as the source of halogen. There was also a possibility that the non-target formula identification software provided wrong results as SV-TAG does not have high resolution mass spectrometer. We added a few more description about it in the manuscript.

*Line 143: It should be noted that the identification of these chemical species was conducted using the unit-mass resolution mass spectrometer of the SV-TAG. Future employment of the high-resolution mass spectrometer will be needed for accurately identifying these chemical species that contain heteroatoms.*

8. Line 135: Where is tygon tubing used in your setup? Can you comment on any particle losses to the tygon tubing? Do you have any blank/background samples to characterize the effect of chemical components (like the plasticizers you mention) coming off the tubing, or chemical compounds of interest sticking to the tubing?

We thank the reviewer for these questions. We mainly used metal (stainless steel or copper) tubing for particle flow. Short Tygon tubings (shorter than 0.5 m in total) were used to connect metal tubings as they were more flexible. Particle loss could be much less than 5% for particles with the diameters of larger than 100 nm in Tygon tubings as demonstrated by a previous study (https://www.tandfonline.com/doi/full/10.1080/15459624.2015.1019077).

We measured the background signal for zero air used in this experiment, and show the corresponding TIC figure in Figure S3. The background sample did not contain the fatty acids of interest, although signals for contaminations such as phthalates were high.

*Line 82: Stainless steel, copper and conductive polytetrafluoroethylene (PTFE) tubings*

*were employed for particle flows. Short pieces of Tygon tubings were also used for connecting them. PTFE tubing was employed both for zero air and ozone flows.*

*Line 138: Chromatograms for all types of COA are shown in Fig. S3, in addition to the corresponding background data.*

*Line 145: Peaks of plasticizers such as phthalate were also identified. The plasticizers were likely originated from contaminants in the zero air in addition to the plastic tubings, as they existed in background measurement (Fig. S3).*

9.  Is section 3.1 all room temperature in the flow tube?

We thank the reviewer for this question. Both sections 3.1 and 3.2 are the results for room-temperature experiments. The title of section 3.2 was modified for clarification.

*Line 182: 3.2 Chemical characteristics of oxidized COA particles at room temperature*

10. Figure 4: It is nice to see the comparison here between the COA types and their signals pre- and post-ozone introduction. I am not surprised to see the spectra looking so similar, since we are getting significant compound fragmentation in the ACSM. I think having a version of this figure with SV-TAG data, similar to Figure 2, would potentially yield more useful chemical information for readers. For instance, you could bring Figure S3 over from the SI and maybe include an inset that zooms in to some peaks of interest that react away significantly with ozone.

We appreciate the reviewer for this suggestion. We have moved the corresponding figure for SV-TAG measurement to the main text and added key compounds related to chemical reactions in the figure.

11. Line 163-164: Can you comment more specifically on what the differences are with the hot pot COA relative to the other types?

We thank the reviewer for this comment. We mentioned the major differences between hot pot and the other two types of particles as below.

*Line 163: Distinct signals from β, γ, and δ tocopherols were found in three types of COA particles, especially in hot pot particles.*

*Line 166: Capsaicin was non-negligible for only hot pot COA particles, likely originating from spices.*

12. Line 234: What are your criteria for these transitions? What does "more pronounced" mean, quantitatively?

We thank the reviewer for this comment. We provided the definition of 'transition temperature' in the revised version. Then we deleted the description of "more pronounced" in the revised version as we have defined the meaning of transition point quantitatively in the previous sentence.

*Line 258: The transition temperature was defined as the point at which $k_2$ became an order of magnitude smaller than that at room temperature.*

13. Line 245: Can you take the fatty acid composition you measured in Figure 3 and estimate the glass transition temperature of each of the mixtures? For instance, as in this paper: https://doi.org/10.5194/acp-18-6331-2018

We appreciate the reviewer for this comment. We agree with the reviewer that estimating the glass transition temperature could help interpret our data. The paper by DeRieux et al. provided equations for transition temperature

estimation based on elemental composition. In the present study, we did not inject any standards for calibrating mass concentrations, except for the major four fatty acids (section 3.1). The lack of quantitative data makes the employment of the equations provided by DeRieux et al. to be challenging. Estimation of glass transition temperature of these mixtures with other techniques (such as ESI and APPI) would be an interesting and important topic in the future. We added the corresponding information in the revised manuscript.

*Line 270: Previous studies demonstrated that numerous processes such as gas and bulk phase diffusion, adsorption and desorption, surface and bulk reaction are involved in determining oxidation rate of aerosol particles (Berkemeier et al., 2021; Pöschl et al., 2007; Li and Knopf, 2021; Willis and Wilson, 2022). Especially, accurate estimation of viscosity is important. A few methods were established to estimate the viscosities and glass transition points of organic compounds based on the chemical compositions and elemental ratios (Derieux et al., 2018; Ceriani et al., 2007). Simultaneous measurements of chemical composition and reactivity will be needed for understanding and estimating chemical aging time scale using molecular-level information of COA particles.*

14. Figure S5: The positive slope for the -11C experiment seems like an anomaly. You mentioned that these data were not used to calculate $k_2$. Can you comment on if these experiments were repeated at all? Did the positive slope for this temperature occur every time? I suggest including information about replicate experiments in the Methods section or maybe even with Table S1.

We conducted replicate experiments for oleic acid particles at room temperature (~25 °C), and the results were consistent. More low temperature experiments were done for canola oil particles (-10 °C, -15 °C, and -20 °C), The fitting slopes for these experiments were close to zero or larger than zero. The trend in $k_2$ values as a function of temperature for the whole range was systematic. Therefore, we treat the chemical reaction transition temperature was somewhere around -15 °C. Based on these two data sets, we considered the reproducibility of our experiments was reasonable.

*Line 247: Figure 6 summarizes the obtained values of $k_2$ plotted against temperature. In general, $k_2$ decreased systematically for lower temperatures.*